

# Response of nine triticale genotypes to different salt concentrations at the germination and early seedling stages

Ebrahim Ramadan[1,*], Haytham A. Freeg[1], Nagwa Shalaby[1], Mosa S. Rizk[1], Jun Ma[2], Wenhua Du[2], Omar M. Ibrahim[3], Khairiah M. Alwutayd[4], Hamada AbdElgawad[5], Ick-Hyun Jo[6] and Amira M. El-Tahan[3,*]

[1] Field Crops Research Institute, Agricultural Research Center, Egypt, Kafr Elshiekh, Egypt
[2] College of Grassland Science, Gansu Agricultural University, Gansu, China
[3] Plant Production Department, Arid Lands Cultivation Research Institute, City of Scientific Research and Technological Applications, Borg El Arab, Alexandria, Egypt
[4] Department of Biology, College of Science, Princess Nourah bint Abdulrahman University, Riyadh, Saudi Arabia
[5] Department of Botany and Microbiology, Faculty of Science, Beni-Suef University, Beni Suef, Egypt
[6] Department of Crop Science and Biotechnology, Dankook University, Cheonan, Republic of Korea
* These authors contributed equally to this work.

Corresponding authors
Ick-Hyun Jo, intron@dankook.ac.kr
Amira M. El-Tahan,
aeltahan@srtacity.sci.eg

## ABSTRACT

Salinity stress poses a major challenge to agricultural productivity worldwide, and understanding their responses at the early growth stage is vital for devising strategies to cope with this stress. Therefore, to improve triticale productivity, this study investigated the salinity stress tolerance of different salt-tolerant triticale genotypes aiming to cultivate them on saline soils. To this end, salinity stress impacts on nine triticale genotypes, *i.e.*, Zhongsi 1084, Gannong No. 2, Gannong No. 4, Shida No. 1, C6, C16, C23, C25 and C36 at germination and early seedling stages was evaluated. Each genotype was subjected to six treatments inducing control, 40, 80, 120, 160, and 200 mM NaCl treatments to study their effect on seedling and termination traits of the nine genotypes. Compared to the overall mean seedling vigor index, the seedling vigor index was higher in the genotypes Zhongsi 1084 and C6 (39% and 18.1%, respectively) and lower in Gannong No.2 (41%). Increasing NaCl concentrations negatively affected germination and seedling traits. Compared to other genotypes, Zhongsi 1084 had the highest mean germination rate, germination vigor index, germination percentage, mean daily germination and germination energy. It also showed the lowest relative salt injury. The relative salt injury was higher in the genotype Shida No. 1 than those in Gannong No. 2, Gannong No. 4, Shida No. 1, C16, and C36 genotypes. All genotypes exhibited desirable mean germination time except for line C6. High significant positive correlations were observed among germination rate, germination vigor index, germination percentage, mean daily germination, seedling vigor index, and root length. Principal component analysis (PCA) grouped the most desirable genotypes into two clusters. Our study determined salt stress tolerance of nine triticale genotypes at germination and early seedling stages. to select salt-tolerant genotypes that can be cultivated on saline soil or after salt irrigation.

# INTRODUCTION

Cereals are an important source of food for both human and animal consumption and nutrition (*Barati & Bijanzadeh, 2021*). Among cereals, triticale (*Triticosecale* Wittmack) is an important cereal crop that belongs to the grass family Poaceae and was developed by hybridizing wheat (*Triticum* spp.) and rye (*Secale cereale*) (*Mohammadi Alagoz et al., 2023*). Two types of triticale have been developed, *i.e.*, hexaploid and octoploid (*Kang et al., 2016*; *Alatrash et al., 2022*). Triticale is rich in protein (*Cantale et al., 2016*; *Biel, Kazimierska & Bashutska, 2020*). Therefore, it is a good food source and feed for cattle, particularly in grazed, stored forage, silage and green fodder (*Zhao et al., 2022*).

The increase in population and the reduction in arable-land area are the two major threats to agricultural sustainability (*Shahbaz & Ashraf, 2013*). In this context, the global population is estimated to be more than 9.8 billion in 2050 (*United Nations, 2023*). Thus, the food demand will be more than double the crop production (*Van Dijk et al., 2021*). To this end, using important crops such as triticale in crop rotation will help minimize soil pests, reduce nutrient levels through leaching and increase crop production (*Cao et al., 2022*). Additionally, the widespread triticale root system contributes to the grain's soil-particle-binding effect (*Demirbas & Balkan, 2020*).

Abiotic stresses are the most significant factors limiting crop development and productivity (*Zhao et al., 2020*). Salinity stress represents the most serious threat to agricultural production, particularly in arid and semi-arid regions where soil nutrient and organic matter levels contribute to physical instability (*Zhao et al., 2020*). It affects approximately one billion hectares of global land worldwide, thus affecting crop production (*Saade et al., 2016*). Furthermore, increasing salinity stress negatively affects all traits of plants associated with germination and early seedling growth. Salinity-induced toxic ions like $Na^+$ and $Cl^-$ affect seed germination by changing osmotic potential, lowering water uptake, causing embryonic damage, and reducing seed germination, shoot elongation, and plant growth (*Farooq et al., 2015*; *Kumar et al., 2020*; *Ullah, Bano & Khan, 2021*). Approximately 20% of the total cultivated area and 33% of irrigated agricultural regions of the world are affected by salinity. Furthermore, the salinized areas are increasing by a rate of 10% annually for several reasons, including low precipitation, high evaporation, irrigation using saline water, and poor cultural practices. Moreover, approximately 50% of arable land will probably be salinized by 2050 (*Jamil et al., 2011*; *Barati & Bijanzadeh, 2021*). In arid and semi-arid regions, salinity is one of the most important environmental factors affecting uniformity in seed germination (*Deng et al., 2020*). Germination is a crucial stage in the development of a plant, as it influences the early growth of the seedling and its relationship with the environment and its productivity (*Mbarki et al., 2020*). Salinity stress induced plant growth inhibition dependent on salt concentration and duration of exposure (*Guo et al., 2022*). It reduces germination rate and capacity of glycophytes (*Saddiq et al., 2021*). This may explained by the increase in osmotic pressure of the soil solution (*Ma et al., 2022*). During germination, the effects of salinity can manifest as osmotic (reversible) and deleterious (irreversible) effects (*Mbarki et al., 2020*). For the most majority of crops, seeds are the means by which sophisticated genetics are transferred to the production

field. Specifically, rapid and synchronous seed germination and seedling growth are vital for the development of seedlings in the field and thus are crucial to crops production (*Reed, Bradford & Khanday, 2022*). Seed germination determines seedling vigor and plant growth. Therefore, this stage is considered a susceptible stage for plant growth (*Hakim et al., 2010*). Improvement in plant growth and establishment in saline soil are dependent on the salt-tolerating ability of the cultivated genotypes in early growth stages (*Keshavarizi & Mohammed, 2012*).

Resilience to abiotic stresses is the driving force behind the development of high-yielding and stable triticale cultivars, which in turn led to an increase in the amount of land used for triticale farming (*Zhao et al., 2020*). Compared to winter cereals, triticale can outproduce on low fertility soils. It has a more robust root system than wheat, barley or oats, allowing it to bond light soils and extract more nutrients (*Saddiq et al., 2021*). Additionally, triticale is tolerant of low pH (acidic soils), sodic soils and boron-rich soils.

Triticale is also a moderate halophyte with high salinity threshold and it is considered a salt-tolerant species (*Grieve, Grattan & Maas, 2012*). It showed salinity tolerant even up to 10 dSm$^{-1}$ (*Ozturk et al., 2018*). The salinity threshold of triticale EC (6.1 dSm$^{-1}$) is higher than that of corn (2.7 dSm$^{-1}$), rye (5.9 dSm$^{-1}$) and wheat (4.7 dSm$^{-1}$). Moreover, (*Kankarla et al., 2020*)reported that the salinity threshold differed among various triticale species compared to other cereals. However, the relative grain yield of triticale genotypes varies at 7.3 dSm$^{-1}$ soil salinity. Each unit increase in soil salinity above 7.3 dSm$^{-1}$ reduced triticale grain yield by 2.8%, placing triticale in the salt-tolerant category (*Francois et al., 1988*).

The establishment of salt-tolerant plants is still in its infancy and shedding the light on the of salinity tolerance mechanisms. Numerous plant species, varieties and halophytes have been studied for their salt tolerance mechanisms, which have proved to be complex (*Mbarki et al., 2020*). Utilizing more appropriate plant cultivars should increase productivity in salinity stressed marginal areas (*Cao et al., 2022*). Thus, for the future of agriculture in arid and semiarid regions, genotypes selection with higher salt tolerance has become an absolute necessity (*Golebiowska-Paluch & Dyda, 2023*). Although triticale is considered as salinity tolerant crop, some genotypes are less tolerant at the germination stage particularly, after the three-leaf growth stage (*Kankarla et al., 2020*). The available literature lacks sufficient information regarding the salinity tolerance of target triticale genotypes (genotypes, *i.e.,* Zhongsi 1084, Gannong No. 2, Gannong No. 4, Shida No. 1, C6, C16, C23, C25 and C36). Hence, our primary goal was to assess the salt stress tolerance of the nine triticale genotypes during the germination and early seedling phases. The overarching aim was to identify genotypes with a high degree of salt tolerance, making them suitable for cultivation in saline soil. This will also provide valuable traits for incorporation into future breeding programs.

## MATERIAL AND METHODS

### Plant genotypes and characteristics

Nine triticale genotypes were used in the current study, and their names and characteristics are listed in Table 1. "Zhongsi 1084", 'Gannong No. 2', 'Gannong No. 4' "Shida 1" and

**Table 1  List of genotypes and names, of triticale investigated in this study.**

| Number | Genotype names |
|--------|----------------|
| 1 | Zhongsi 1084 (Chinese Triticale cultivar) |
| 2 | Gannong No. 2 (Chinese Triticale cultivar) |
| 3 | Gannong No. 4 (Chinese Triticale cultivar) |
| 4 | Shida No. 1 (Chinese Triticale cultivar) |
| 5 | C6 (Triticale line bred by GASU) |
| 6 | C16 (Triticale line bred by GASU) |
| 7 | C23 (Triticale line bred by GASU) |
| 8 | C25 (Triticale line bred by GASU) |
| 9 | C36 (Triticale line bred by GASU) |

Notes.
GASU, Gansu Agricultural University of P.R. China.

lines "C6, C16, C23, C25, C36" were bred by the College of Grassland Science, Gansu Agricultural University, China, using the traditional sexual hybridization techniques and a pedigree selection method (*Ramadan et al., 2023*).

## Study location

The experiment was conducted at Gansu Agricultural University, P. R. China. 36°5′26″ north, 103°41′41″ east.

## Germination conditions

The seeds of the studied genotypes were sterilized using sodium hypochlorite (1%) for 30 min and washed thrice using distilled water. Next, 50 seeds of each genotype were germinated on Whatman No. 1 filter paper in 9-cm Petri dishes under the following six NaCl concentrations: control, 40 mM, 80 mM, 120 mM, 160 mM, and 200 mM. The seeds were allowed to germinate in an incubator at $20 \pm 1$ °C under a $16/8-h$ dark/light cycle for 7 d (*Warham, Butler & Sutton, 1995*); they were irrigated and washed twice daily using their corresponding treatment solution, and the filter papers were changed once every 2 d to prevent salt accumulation. After 2 d of planting, the germinated seeds were counted; the seeds were considered to have been germinated when the emerging radicle was one mm in length. Germination percentage was evaluated every 24 h for 5 d.

## Analysis of different germination and growth parameters

After 7 d of planting, shoot length (SL; cm), root length (RL; cm), shoot fresh weight (SFW; mg), root fresh weight (RFW; mg), shoot dry weight (SDW; mg), root dry weight (RDW; mg), and root/shoot dry weight ratio (RSR) were measured. Dry weight was measured after drying the roots or shoots at 70 °C for 72 h in an oven.

Germination traits were calculated as follows:

$$\textbf{Germination rate (GR)} = \sum_{i=1}^{n} S_i/D_i \; (\textit{Maguire, 1962}). \tag{1}$$

$S_i$ is the germinated seeds per total seeds, $D_i$ represents seed numbers until $n$thday, and n is the number of counting.

$$\textbf{Germination vigor index (GVI)} = \sum_{i=1}^{k} n_i/t_i \text{ (\textit{Maguire, 1962}).} \tag{2}$$

$n_i$ is the percentage of seeds germinated on the $n$th day, and $t_i$ is the number of days counted from the start of the experiment ($i$) to the last day on which the seeds germinated ($k$). Higher values represent a more rapid rate of germination.

$$\textbf{Germination percentage (GP}\%) = \text{(Seeds germinated/Total seeds)}$$
$$\times \; 100 \text{ (\textit{Manmathan \& Lapitan, 2013}).} \tag{3}$$

$$\textbf{Mean daily germination (MDG)} = \text{Final germination percentage}$$
$$\text{/number of days to final germination} \tag{4}$$

$$\textbf{Mean germination time (MGT)} = \sum(T_i N_i)/\sum N_i \text{ (\textit{Kankarla et al., 2020}).} \tag{5}$$

$N_i$ is the number of the newly germinated seeds in times of $T_i$

$$\text{The } \textbf{energy of germination (GE)} = \text{Percentage of the germinated seeds 4 d after planting}$$
$$\text{/Total number of seeds tested (\textit{Ruan, Xue \& Tylkowska, 2002}).} \tag{6}$$

$$\textbf{Relative salt injury (RSI)} = \text{(Germination percentage of the control} - \text{Germination}$$
$$\text{percentage of the treatment)/Germination percentage of the control} \tag{7}$$

$$\textbf{Seedling vigor index (SVI)} = \text{(Average shoot length} + \text{Average root length)}$$
$$\times \; \text{Germination percentage (\textit{Abdul-Baki \& Anderson, 1973}).} \tag{8}$$

## Salinity stress tolerance

As a quantitative measure, stress indices can quantify the stress responses of a crop. They are easier to use and interpret than raw data. Many indices of abiotic stress tolerance have been proposed (Table 2) for estimating abiotic stress tolerant genotypes using a mathematical equation that describes the relationship between growth under stress and control conditions. The abiotic stress indices are classified into two types: The first type contains indices with maximum values indicating high-stress tolerance, whereas the other type includes other indices with minimum values indicating high-stress tolerance. Using these indices, the tolerant and sensitive genotypes and their stability can be identified (*Parvaze & Ahmad, 2018*).

## Statistical analysis

The experiment was performed as per a factorial, completely randomized design (CRD) (where Factor-1 was genotype including nine levels, and Factor-2 was salt stress treatments including six levels) with three replicates and 50 seeds in each replicate. Two-way analysis of variance (ANOVA) was used for data analysis using SAS statistical software, version 9.2 (SAS Institute, Cary, NC, USA). The means were compared using Duncan's multiple range

**Table 2  Abiotic stress screening indices.**

| Index | Formula | Reference |
|---|---|---|
| Indices with maximum values corresponding to more tolerant | | |
| Mean productivity (MP) | $(Y_S + Y_{NS})/2$ | *Rosielle & Hamblin (1981)* |
| Geometric mean productivity (GMP) | $(Y_{NS})^{(1/2)} \times Y_S$ | *Fernandez (1992)* |
| Harmonic mean (HM) | $2 \times (Y_S \times Y_{NS})/(Y_S + Y_{NS})$ | *Bidinger, Mahalakshmi & Rao (1987)* |
| Stress Tolerance Index (STI) | $(Y_S \times Y_{NS})/(Y_{NS.m})^2$ | *Fernandez (1992)* |
| Yield index (YI) | $Y_s/Y_{S.m}$ | *Gavuzzi et al. (1997)* |
| Modified stress tolerance index-I (MSTI1) | $((Y_{NS})^2/(Y_{NS.m})^2) \times ((Y_s \times Y_{NS})/(Y_{NS.m})^2)$ | *Farshadfar & Sutka (2003)* |
| Modified stress tolerance index- II (MSTI2) | $((Y_s)^2/(Y_{S.m})^2) \times ((Y_s \times Y_{NS})/(Y_{NS.m})^2)$ | *Farshadfar & Sutka (2003)* |
| Yield stability index (YSI) | $Y_s/Y_{NS}$ | *Bouslama & Schapaugh (1984)* |
| Relative stress index (RSI) | $(Y_s/Y_{NS})/(Y_{S.m}/Y_{NS.m})$ | *Fischer & Wood (1979)* |
| Drought index (DI) | $(Y_s * (Y_s/Y_{NS}))/Y_{S.m}$ | *Bidinger, Mahalakshmi & Rao (1987)* |
| Stress/non-stress productivity index (SNPI) | $((Y_{NS} + Y_s)/(Y_{NS} - Y_s))^{(1/3)} \times (Y_{NS} \times Y_s \times Y_s)^{(1/3)}$ | *Moosavi et al. (2008)* |
| Relative efficiency index (REI) | $(Y_s \times Y_{NS})/(Y_{S.m} \times Y_{NS.m})$ | *Ramirez-Vallejo & Kelly (1998)* |
| Mean relative performance (MRP) | $(Y_s/Y_{S.m}) + (Y_{NS}/Y_{NS.m})$ | *Ramirez-Vallejo & Kelly (1998)* |
| Golden mean (Gm) | $(Y_{NS} + Y_s)/(Y_{NS} - Y_s)$ | *Moradi et al. (2012)* |
| Indices with minimum values corresponding to more tolerant genotype | | |
| Tolerance index (TOL) | $Y_{NS} - Y_s$ | *Rosielle & Hamblin (1981)* |
| Stress susceptibility Index (SSI) | $(1 - (Y_s/Y_{NS}))/(1 - (Y_{S.m}/Y_{NS.m}))$ | *Schneider et al. (1997)* |
| Stress susceptibility percentage index (SSPI) | $(Y_{NS} - Ys)/(2 \times Y_{NS.m})$ | *Moosavi et al. (2008)* |
| Yield reduction (YR) | $1- (Ys/Y_{NS})$ | *Choukan et al. (2006)* |
| Abiotic stress tolerance index (ATI) | $((Y_{NS} - Ys)/(Y_{NS.m}/Y_{S.m})) \times (Y_{NS} \times Ys)^{(1/2)}$ | *Moosavi et al. (2008)* |
| Mean productivity index (MPI) | $(Y_{NS} - Ys)/2$ | *Rosielle & Hamblin (1981)* |
| Schnieder's stress susceptibility index (SSSI) | $1 - (Ys/Y_{NS}) - (1 - (Y_{S.m}/Y_{NS.m}))$ | *Schneider et al. (1997)* |
| Sensitivity drought index (SDI) | $(Y_{NS} - Ys)/Y_{NS}$ | *Farshadfar & Javadinia (2011)* |

test ($P < 0.05$), and correlation coefficient was calculated using SPSS version 16. Principal component analysis (PCA) was performed using the statistical package PAST (*Hammer, Harper & Ryan, 2001*) to visualize the differences in various stress-related traits among the nine genotypes.

To categorize the genotypes under both control and salinity stress treatments, cluster analysis was performed using R software version 4.1.0, 2021 (*R Core Team, 2021*). Euclidian metric as a distance measure was used to measure dissimilarity among the genotypes, and *Ward Jr.'s (1963)* algorithm was applied for grouping the genotypes.

Before conducting the analysis, the data were standardized due to their different scale by subtracting the mean from each value and dividing the obtained value by the standard deviation. The cubic cluster criterion (*Milligan & Cooper, 1985*) was used to ensure whether clusters existed. Fuzzy C-means as a soft clustering algorithm (*Bezdek, 1973*; *Bezdek, 1981*) was used to detect if overlapping existed between clusters. PCA is a multi-variable statistical analysis that reduces the dimensions of high-dimension data, and fewer eigenvectors explain the multivariate data (*Shlens, 2005*).

**Table 3  Mean square estimates for the parameters of triticale genotypes under different salt treatments.**

| Source of Variance | Treatment | Genotype | Treatment × Genotype | Error |
|---|---|---|---|---|
| Degree of fredom | 5 | 8 | 40 | 108 |
| Germination rate | 47.51** | 8.59** | 0.365** | 0.06 |
| Germination vigor index | 2891.46** | 470.32** | 18.675** | 4.16 |
| Germination percentage (%) | 18758.28** | 3197.63** | 170.011** | 29.56 |
| Mean daily germination | 382.77** | 65.25** | 3.475** | 0.60 |
| Mean germination time (d) | 7.67** | 0.48$^{ns}$ | 0.413* | 0.27 |
| Germination energy (%) | 1164.17** | 168.13** | 55.973** | 22.18 |
| Relative salt injury | 25066.84** | 2360.53** | 243.20** | 43.22 |
| Seedling vigor index | 881.77** | 30.83** | 4.481** | 0.58 |
| Shoot length (cm) | 268.10** | 3.44** | 1.43** | 0.29 |
| Root length (cm) | 152.64** | 1.90** | 0.67** | 0.18 |
| Root/shoot ratio | 0.54** | 0.02** | 0.01** | 0.01 |
| Shoot fresh weight (mg) | 382627.79** | 11844.21** | 2341.06** | 911.49 |
| Root fresh weight (mg) | 88069.30** | 6957.79** | 1506.71** | 515.80 |
| Shoot dry weight (mg) | 5271.16** | 171.06** | 38.94** | 13.39 |
| Root dry weight (mg) | 1217.71** | 143.46** | 18.28** | 7.11 |

**Notes.**
*Significant differences at the 0.05 level.
**Highly significant differences at 0.01 level.
ns, no significant differences.

## RESULTS

### Genotypes differentially responded to salinity stress

To study genotype specific responses to salt stress treatment, Analysis of variance (ANOVA) were performed. Highly significant mean squares due to the genotypes and treatments, and genotypes × treatments were detected for all studied traits except for MGT, where the mean square was non-significant for genotypes and significant for the interaction between treatments and genotypes (Table 3). This result indicated high variation among the studied genotypes under different salt stress treatments.

### Mean performance of different genotypes

Analysis of germination traits (Table 4) revealed that the triticale genotype Zhongsi 1084 had the highest mean GR, GVI, GP%, MDG and GE. In contrast, the genotypes Gannong No.2 and Shida No. 1 exhibited the lowest mean of GR, GVI, GP% and MDG. There was no significant difference between the two cultivars. All genotypes exhibited the highest MGT except for line C6. The lowest RSI was observed for Zhongsi 1084, whereas the highest RSI was observed in case of Shida No.1. The SVI of genotypes Zhongsi 1084 and C6 was 39% and 18.1%, respectively. This was higher than the overall mean SVI, whereas that of Gannong No.2 was 41% less than this mean.

Analysis of seedling traits revealed that the mean SL of C6 and Zhongsi 1084 was 12.4% and 9.1% higher than the overall mean of SL, respectively. Whereas the means SL of Gannong No. 2 and C16 were 16.7% and 6.1% lower than the overall mean of SL,

Ramadan et al. (2023), *PeerJ*, DOI 10.7717/peerj.16256

**Table 4** The overall mean performance of different studied triticale genotypes under six salt treatments.

| Genotypes<br>Traits | | Zhongsi<br>1084 | Gannong<br>No. 2 | Gannong<br>No. 4 | Shida<br>No. 1 | C6 | C16 | C23 | C25 | C36 | Mean |
|---|---|---|---|---|---|---|---|---|---|---|---|
| | Germination rate | 3.93[a] | 1.85[e] | 2.89[c] | 1.91[e] | 3.26[b] | 2.40[d] | 3.14[b] | 3.19[b] | 2.33[d] | 2.77 |
| | Germin. vigor index | 28.83[a] | 13.76[f] | 22.02[d] | 14.64[f] | 24.98[b] | 17.50[e] | 23.30[cd] | 23.92[bc] | 17.44[e] | 20.71 |
| | Germin. (%) | 79.15[a] | 38.49[f] | 56.65[c] | 39.06[f] | 63.32[b] | 49.52[d] | 63.10[b] | 62.64[b] | 45.74[e] | 55.3 |
| | Mean daily germin. | 11.31[a] | 5.50[f] | 8.09[c] | 5.58[f] | 9.05[b] | 7.07[d] | 9.01[b] | 8.95[b] | 6.53[e] | 7.9 |
| | Mean germin. time<br>(days) | 3.14[ab] | 3.26[a] | 2.94[ab] | 3.2[ab] | 2.85[b] | 3.29[a] | 3.28[a] | 2.96[ab] | 3.05[ab] | 3.11 |
| | Germin. energy | 49.26[a] | 42.66[b] | 43.78[b] | 45.43[b] | 49.09[a] | 39.42[c] | 44.62[b] | 46.17[ab] | 44.93[b] | 45.04 |
| | Relative salt injury | 0.19[g] | 0.50[c] | 0.51[c] | 0.64[a] | 0.31[f] | 0.47[cd] | 0.38[e] | 0.43[d] | 0.58[b] | 0.45 |
| | Seedling vigor index | 7.57[a] | 3.22[e] | 6.03[bc] | 4.68[d] | 6.44[b] | 4.59[d] | 6.13[bc] | 5.85[c] | 4.51[d] | 5.45 |
| | Shoot length (cm) | 5.42[ab] | 4.14[f] | 5.27[abc] | 5.16[bcd] | 5.58[a] | 4.67[e] | 4.97[cde] | 4.80[de] | 4.85[de] | 4.99 |
| Germination<br>traits | Root length (cm) | 3.11[a] | 2.31[de] | 2.86[ab] | 2.55[cd] | 3.07[ab] | 2.46[cde] | 2.94[ab] | 2.75[bc] | 2.20[e] | 2.69 |
| | Root/shoot ratio | 0.48[ab] | 0.51[a] | 0.47[ab] | 0.48[ab] | 0.45[b] | 0.45[b] | 0.51[a] | 0.49[ab] | 0.40[c] | 0.47 |
| | Shoot fresh weight<br>(mg) | 258.99[c] | 223.25[e] | 287.44[ab] | 271.74[bc] | 297.25[a] | 230.84[de] | 251.52[cd] | 231.62[de] | 255.94[c] | 256.51 |
| | Root fresh weight<br>(mg) | 122.03[bc] | 114.36[bc] | 148.09[a] | 124.95[b] | 162.56[a] | 108.62[bc] | 111.40[bc] | 105.83[c] | 109.49[bc] | 123.04 |
| | Shoot dry weight<br>(mg) | 33.53[b] | 29.65[c] | 37.18[a] | 37.78[a] | 38.52[a] | 31.32[bc] | 33.33[b] | 31.86[bc] | 33.99[b] | 34.13 |
| | Root dry weight<br>(mg) | 20.57[b] | 17.32[c] | 23.48[a] | 20.25[b] | 24.26[a] | 17.55[c] | 16.85[c] | 17.04[c] | 18.10[c] | 19.49 |

**Notes.**

Values followed by the different letter(s) are significantly different from each other by Duncan's multiple range test at 5% level of probability.

respectively. Moreover, the mean RL of Zhongsi 1084, C6, and C23 was the highest. They recorded 17.8%, 16.2%, and 11.3% higher than the overall mean value, respectively. The mean RL of genotypes C36 and Gannong No. 2 was the lowest *i.e.,* 16.8% and 12.4% lesser than the overall mean value, respectively. The highest RSRs were observed for genotypes C23 and Gannong No. 2, whereas the lowest ratio was observed for genotype C36. The highest increase in SFW compared with the overall mean SFW was observed for genotypes C6 (15.9%) and Gannong No. 4 (12.1%). Whereas the highest decrease was observed in Gannong No. 2 (13%), C16 (10%), and C25 (9.7%). For RFW, the highest mean values were exhibited by C6 and Gannong No. 4. They recorded 32.1% and 20.4% more than the general mean, respectively. Meanwhile, genotypes C25, C16 and C36 exhibited the lowest mean values *i.e.,* they were 13.1%, 11.7%, and 11% lower than the overall mean, respectively. Genotype C6 had the highest mean SDW *i.e.,* 12.9% higher than the general mean. Meanwhile, both genotypes Gannong No. 2 and C16 had the lowest mean SDW values, exhibiting 13.1% and 8.2% decreases compared to the general mean, respectively. Moreover, both genotypes C6 and Gannong No. 4 had the highest mean RDW values. They were 24.5% and 20.5% higher than the general mean. The mean RDW values of genotypes C23, C25 and Gannong No. 2 were 13.5%, 12.6%, and 11.1% was lower than the general mean.

## Differential effect of salt treatments on germination

The GR of different triticale genotypes under varying salt concentrations were 1.42–4.4% (Table 5). The highest GR was observed in the control and 40 mM NaCl treated groups. GR gradually also reduced with increasing NaCl concentrations. GR reduced by 41% and 67.9% in the 80 mM and 200 mM NaCl treated groups, respectively. GVI was significantly different under different salinity levels, whereas mean values of different treatments were 10.28–33.54. The highest value was observed in the 40 mM NaCl treated group *i.e.,* the lowest value was observed in the 200 mM NaCl group. Moreover, significant differences were not observed between the control and 40 mM NaCl groups. The highest reduction in GVI was observed, where 60%, 62%, and 69.2 were observed under 120, 160, and 200 mM NaCl treatments, respectively. However, the significant differences were not observed in GP% between control and 40 mM NaCl treated groups. In contrast, significant differences in GP% were observed as NaCl concentration increased from 80 mM to 200 mM and the GP% reduced by 39.8% in the 80 mM NaCl group. The highest GP% (88.04%) was observed for the 40 mM NaCl treated group, whereas the lowest GP% (28.29%) was observed in the 200 mM NaCl treated group. The highest MDG was observed in both control and 40 mM NaCl groups (12.50 and 12.58, respectively) and the lowest value was observed in the 200 mM NaCl treated group. The reduction % in MDG increased from 39.8% to 67.4% as NaCl concentration increased from 80 mM to 200 mM. The number of days required for germination increased from 2.48 day in the control group to 4.09 day in the 120 mM NaCl treated group. In groups treated with NaCl concentration >120 mM, the number of days for germination gradually decreased with increasing NaCl concentrations. However, significant differences were not observed among 40, 80, 160, and 200 mM NaCl groups. GE decreased from 48.76% in the control group to 35.96% in the 120 mM NaCl

**Table 5  The overall mean performance of the six salt treatments.**

| Treatments Traits | | Control | 40 mM | 80 mM | 120 mM | 160 mM | 200 mM | Mean |
|---|---|---|---|---|---|---|---|---|
| Germination traits | Germination rate | 4.40[a] | 4.40[a] | 2.60[b] | 2.04[c] | 1.74[d] | 1.42[e] | 2.77 |
| | Germin. vigor index | 33.39[a] | 33.54[a] | 19.65[b] | 14.69[c] | 12.71[d] | 10.28[e] | 20.71 |
| | Germin. (%) | 87.50[a] | 88.04[a] | 52.66[b] | 39.63[c] | 35.47[d] | 28.49[e] | 55.30 |
| | Mean daily germin. | 12.50[a] | 12.58[a] | 7.52[b] | 5.66[c] | 5.07[d] | 4.07[e] | 7.90 |
| | Mean germin. time (days) | 2.48[c] | 2.94[b] | 3.15[b] | 4.09[a] | 3.03[b] | 2.96[b] | 3.11 |
| | Germin. energy (%) | 48.76[ab] | 47.61[b] | 37.51[c] | 35.96[c] | 49.05[ab] | 51.35[a] | 45.04 |
| | Relative salt injury | 0.00[e] | −0.62[e] | 39.82[d] | 54.71[c] | 59.47[b] | 67.44[a] | 36.80 |
| | Seedling vigor index | 14.46[a] | 10.53[b] | 3.96[c] | 2.04[d] | 1.06[e] | 0.64[f] | 5.45 |
| | Shoot length (cm) | 9.83[a] | 7.42[b] | 5.14[c] | 3.60[d] | 2.15[e] | 1.77[f] | 4.99 |
| | Root length (cm) | 6.57[a] | 4.44[b] | 2.32[c] | 1.50[d] | 0.85[e] | 0.48[f] | 2.69 |
| | Root / shoot ratio | 0.67[a] | 0.60[b] | 0.45[c] | 0.40[d] | 0.40[d] | 0.30[e] | 0.47 |
| | Shoot fresh weight (mg) | 411.88[a] | 355.74[b] | 298.72[c] | 233.47[d] | 130.17[e] | 101.20[f] | 255.20 |
| | Root fresh weight (mg) | 208.23[a] | 169.93[b] | 128.82[c] | 89.86[d] | 69.42[e] | 64.56[e] | 121.80 |
| | Shoot dry weight (mg) | 51.09[a] | 45.82[b] | 40.50[c] | 30.28[d] | 20.75[e] | 16.32[f] | 34.13 |
| | Root dry weight (mg) | 29.27[a] | 25.04[b] | 20.89[c] | 15.54[d] | 13.20[e] | 13.01[e] | 19.49 |

**Notes.**
Values followed by the different lowercase letter(s) are significantly different from each other by Duncan's multiple range test at 5% level of probability.

group. However, in groups with NaCl concentration > 120 mM, GE gradually increased and it was 51.35% at 200 mM NaCl. However, significant differences in GE were not observed between the control, and 160 mM and 200 mM NaCl treated groups. The RSI was negative in the 40 mM NaCl treated group and increased significantly with increasing salt concentration. It increased from 39.82% under 80 mM NaCl treatment to 67.44% under 200 mM NaCl treatment. SVI decreased with increasing salt concentrations, where SVI was reduced by 27.2% in the 40 mM NaCl treated group and by 95.6% in the 200 mM NaCl treated group.

Both SL and RL reduced significantly with increasing salt stress (Table 5). The highest mean values were observed in the control group, whereas the lowest mean values were recorded in the 200 mM NaCl group. Mean SL varied from 9.83 cm to 1.77 cm, and the reduction in SL ranged from 27.2% (40 mM NaCl) to 82% (200 mM NaCl). Mean RL varied from 6.57 cm to 0.48 cm, and the reduction in RL ranged from 32.4% (40 mM NaCl) to 92.7% (200 mM NaCl). RSR gradually decreased from 0.67 in the control group to 0.3 in the 200 mM NaCl treated group; however, significant differences were not observed between the 120 and 160 mM NaCl treated groups. In the 200 mM NaCl group, >50% reduction in RSR was observed compared to that of the control group. SFW and SDW were significantly affected by salt stress. Compared to that of the control group, reduction in SFW and SDW was 13.6–75.4% and 10.3–68.1% under increased NaCl concentration from 40 mM to 200 mM. Moreover, RFW and RDW were significantly reduced by salinity, where increasing NaCl concentration increased from 40 mM to 200 mM reduced the RFW and RDW values by18.4–69% and 14.5–55.6%, respectively.

## Interaction effects

The mean performance of the different triticale genotypes under salt stress (Figs. 1A, 1B, 2A and 2B). The highest GR, GVI, and GP% were observed for Zhongsi 1084 under 40–200 mM NaCl treatments, whereas the lowest values were observed for Shida No.1 under 80–200 mM NaCl treatments. Zhongsi 1084 exhibited the best MDG under 40–200 mM NaCl treatments, whereas Shida No. 1 was the most affected under high salt concentrations (120–200 mM NaCl). MGT was 2.01–3.41, 2.7–3.31, 2.58–3.96, 3.23–4.59, 2.68–3.53 and 2.59–3.37 days for control, and 40 mM, 80 mM, 120 mM, 160 mM, and 200 mM NaCl treated groups, respectively. The lowest number of days under control and 120 mM NaCl treatments was observed in genotype C6. Gannong No. 4 exhibited the best GE under control treatment (55.97%), Zhongsi 1084 under 40 and 120 mM NaCl treatments (52.84 and 48.03%, respectively), C6 under 80 mM treatment (46.06%) and Shida No. 1 under 160 mM and 200 mM NaCl treatments (56.5 and 57.5%, respectively). RSI increased with increasing salt concentrations. The lowest percentage of injury was observed in Zhongsi 1084 *i.e.,* 10.23, 24.18, 25.36 and 38% under 80, 120, 160, and 200 mM NaCl treatments, respectively. Meanwhile, the highest percentage of injury was observed in Shida No. 1 (57.57, 82.17, 87.38, and 87.21% under 80, 120, 160, and 200 mM NaCl treatments, respectively). For SVI, the most desirable genotypes were Zhongsi 1084 and Gannong No. 4 under control treatment; Zhongsi 1084 and C6 under 40 mM, 120, and 200 mM NaCl treatments; Zhongsi 1084 and C23 under 80 mM NaCl treatments; and both Zhongsi 1084 and C25 under 160 mM NaCl treatments. In contrast, Shida No. 1 was the most affected genotype under high salt concentrations.

Furthermore, Zhongsi 1084 had the highest mean SL under control and 40 mM NaCl treatments, but the lowest mean SL under 160 and 200 mM NaCl treatments. C6 had the mean highest SL under 80 and 120 mM NaCl treatments. C16 also had the highest mean SL under 160 and 200 mM NaCl treatments. The lowest mean SL under control and 40 mM, 80 mM, and 120 mM NaCl treatments were exhibited by Gannong No. 2. The mean RL was 8.57–5.13 and 5.45–3.61 cm for control and 40 mM NaCl groups, respectively. It gradually decreased to 0.97–0.62 and 0.61–0.29 cm for 160 mM and 200 mM NaCl treated groups, respectively. The mean RSR decreased with increasing salt concentrations. The mean ratios ranged from 0.79 to 0.53 under control and from 0.42 to 0.20 under 200 mM NaCl treatment. Shida No. 1 had the highest RSRs under 160 and 200 mM NaCl treatments, whereas C16 had the lowest ratios. Both C6 and Shida No. 1 were the best genotypes as per their mean SFW under 0, 40 and 80 mM NaCl treatments, whereas both Gannong No. 4 and C6 were the best genotypes under 120, 160, 200 mM NaCl treatments as per their mean SFW. C6 and Gannong No. 4 were the best genotypes as per mean RFW under 0-120 mM NaCl treatments, whereas C6 and Gannong No. 2 were the best under 160 and 200 mM NaCl treatments. Furthermore, the highest mean SDW was observed for Shida No. 1 and C6 under 0–80 mM NaCl treatments, for C6 under 120 mM and 160 mM NaCl treatments and for Gannong No. 4 under 200 mM NaCl treatment. In contrast, the lowest mean SDW under high salt concentrations was exhibited by Zhongsi 1084 and C36. Gannong No. 4 and C6 were the most desirable genotypes under 0-120 mM salt treatments for RDW. It was also reported that different salinity concentrations caused considerable

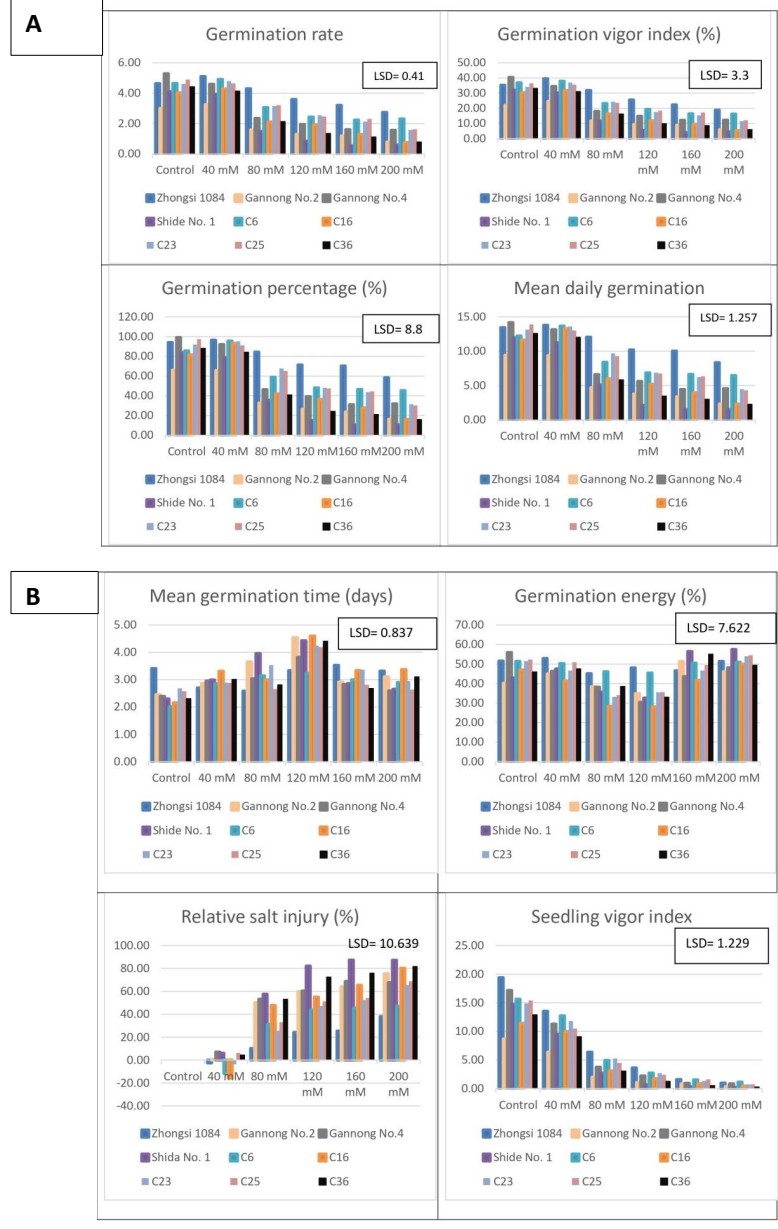

**Figure 1** (A–B) Mean performance of germination traits as affected by the interaction between genotypes and salt treatments (mM NaCl).

effects on GP%, GR, total dry weight, and all seedling traits in all studied genotypes. Similar results for the interaction between salt stress and genotypes have been reported by *Kandil et al. (2012)*.

## Phenotypic correlation

Phenotypic correlation coefficients among the studied traits (Table 6). The highest positive correlation ($r = 1.00$) was observed between GP% and MDG. High significant positive

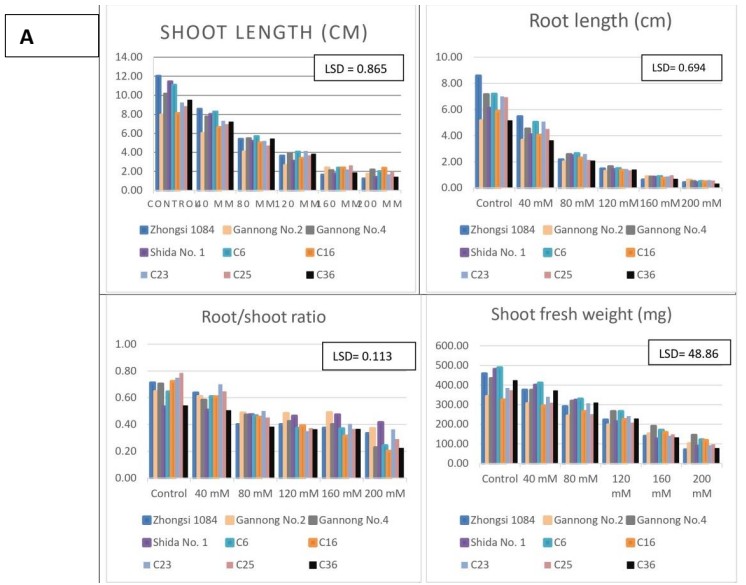

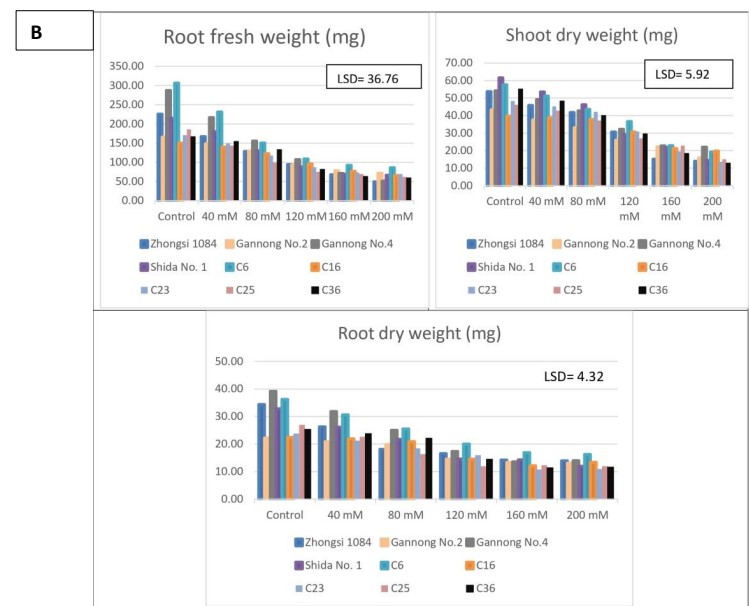

**Figure 2 (A–B) Mean performance of seedling traits as affected by the interaction between genotypes and salt treatments (mM NaCl).**

correlations were observed among GR, GVI, GP%, MDG, SVI, and RL. Significant positive correlations were also observed among RL, SFW, RFW and RDW. SVI was significantly positively correlated with RL. GVI was significantly positively correlated with GE and SL. Significant positive correlations were observed between GE, SVI, and SL, and between SL and RL. Positive but non-significant correlations were observed between germination traits GR, GVI, GP%, MGT, GE and SVI and seedling traits RSR, SFW, RFW, SDW and RDW.

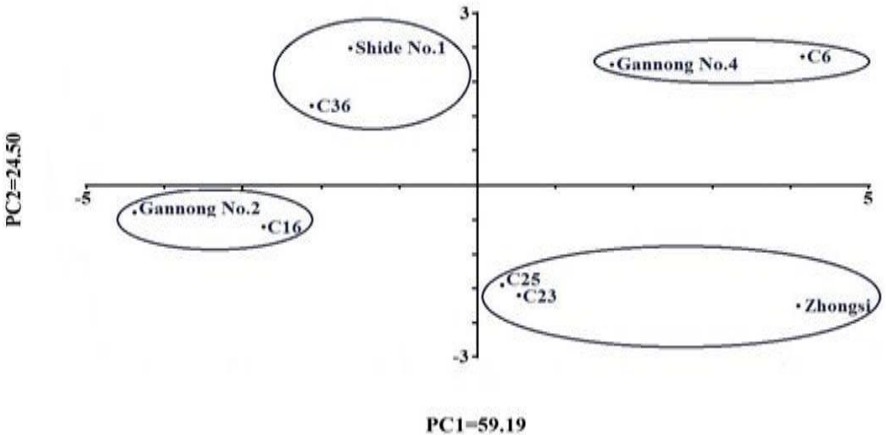

**Figure 3** Two-dimensional ordination of the nine Triticale genotypes investigated in this study based on their overall mean performance under salt treatments.

In contrast, highly significant negative correlations were observed between RSI and GR, GVI, GP%, and MDG. Significant negative correlations were also observed between MGT and RDW and between RSI and both SVI and RL.

## PCA

In the current study, PCA classified the nine genotypes into four clusters based on their mean performance under different NaCl treatments (Fig. 3). The first cluster was found in the 1st quadrant, which included triticale genotypes C6 and Gannong No. 4. Both genotypes scored the highest values for the seedling traits SFW, RFW, SDW, RDW and high values for RL, SVI, MDG and GVI. The second cluster was found in the 2nd quadrant and included the genotypes Zhongsi 1084, C23, and C25. These genotypes had high mean GR, GVI, GP%, MDG, SVI, SL and RL and low RSI. The third cluster was found in the 3rd quadrant and included Gannong No. 2 and C16 genotypes, whereas the fourth cluster was found in the 4th quadrant and included both Shida No. 1 and C36. The genotypes in the third and the fourth clusters had the lowest mean GR, GVI, GP%, MDG, SVI, and RL. These results suggested considerable variability for salt tolerance in the studied triticale genotypes.

Tables 7 and 8 reveal that Gannong No. 4 was the most tolerant genotype with an average rank (AR) equal to 2.12 (Fig. 4). However, Zhongsi 1084 was the least tolerant genotype (AR = 8.04). Both Gannong No. 2 and C25 were moderately tolerant as their ARs were 4.29 and 4.62, respectively. Higher AR suggested the lower tolerance of the genotype (Table 8).

## Cluster analysis

SFW and RFW were used to construct a distance matrix and to generate a tanglegram exhibiting dissimilarity among all genotypes under control and the treatment with the highest salt concentration (200 mM) (Fig. 5). The fuzzy C-means method elucidated

**Table 6  Phenotypic correlation coefficients among the studied traits.**

| Traits | GR | GVI | GP | MDG | MGT | GE | RSI | SVI | SL | RL | RSR | SFW | RFW | SDW |
|---|---|---|---|---|---|---|---|---|---|---|---|---|---|---|
| GVI | 0.997** | | | | | | | | | | | | | |
| GP | 0.996** | 0.988** | | | | | | | | | | | | |
| MDG | 0.996** | 0.988** | 1.000** | | | | | | | | | | | |
| MGT | −0.4 | −0.462 | −0.333 | −0.333 | | | | | | | | | | |
| GEN | 0.652 | 0.681* | 0.600 | 0.600 | −0.569 | | | | | | | | | |
| RSI | −0.881** | −0.864** | −0.900** | −0.900** | 0.200 | −0.566 | | | | | | | | |
| SVI | 0.952** | 0.960** | 0.944** | 0.944** | −0.432 | 0.700* | −0.769* | | | | | | | |
| SL | 0.642 | 0.677* | 0.614 | 0.614 | −0.567 | 0.708* | −0.432 | 0.823** | | | | | | |
| RL | 0.868** | 0.886** | 0.864** | 0.864** | −0.382 | 0.648 | −0.771* | 0.928** | 0.777* | | | | | |
| RSR | 0.203 | 0.200 | 0.240 | 0.240 | 0.330 | 0.060 | −0.272 | 0.200 | −0.100 | 0.430 | | | | |
| SFW | 0.292 | 0.348 | 0.246 | 0.246 | −0.627 | 0.542 | −0.106 | 0.506 | 0.869** | 0.536 | −0.219 | | | |
| RFW | 0.251 | 0.308 | 0.209 | 0.209 | −0.654 | 0.445 | −0.218 | 0.388 | 0.694* | 0.533 | −0.104 | 0.883** | | |
| SDW | 0.156 | 0.214 | 0.109 | 0.109 | −0.572 | 0.492 | 0.065 | 0.409 | 0.835** | 0.445 | −0.199 | 0.963** | 0.798** | |
| RDW | 0.303 | 0.354 | 0.264 | 0.265 | −0.672* | 0.488 | −0.212 | 0.467 | 0.782* | 0.535 | −0.191 | 0.913** | 0.962** | 0.837** |

**Notes.**

Where: GR, germination rat; GVI, germination vigor index; GP, germination percentage; MDG, mean daily germination; MGT, mean germination time; GE, germination energy; RSI, relative salt injury; SVI, seedling vigor index; SL, shoot length; RL, root length; RSR, root/shoot ratio; SFW, shoot fresh weight; RFW, root fresh weight; SDW, shoot dry weight, RDW, root dry weight.

*Significant differences exited at the 0.05 level.

**Highly significant differences exited at the 0.01 level.

**Table 7** Values of 22 abiotic stress indices based on shoot fresh weight under stress (Ys) and control (Yc) treatments.

| | Zhongsi 1084 | Gannong No. 2 | Gannong No. 4 | Shida No. 1 | C6 | C16 | C23 | C25 | C36 |
|---|---|---|---|---|---|---|---|---|---|
| Yield under normal condition (Yns) | 457.67 | 342.25 | 433.00 | 480.33 | 488.33 | 324.67 | 385.00 | 373.33 | 422.33 |
| Yield under stress condition (Ys) | 70.00 | 101.87 | 144.00 | 88.33 | 121.17 | 117.80 | 93.34 | 97.97 | 76.33 |
| Mean productivity (MP) | 263.83 | 222.06 | 288.50 | 284.33 | 304.75 | 221.23 | 239.17 | 235.65 | 249.33 |
| Geometric mean productivity (GMP) | 1497.52 | 1884.53 | 2996.45 | 1935.96 | 2677.57 | 2122.58 | 1831.40 | 1892.90 | 1568.71 |
| Harmonic mean (HM) | 121.43 | 157.00 | 216.12 | 149.22 | 194.16 | 172.88 | 150.25 | 155.21 | 129.30 |
| Stress Tolerance Index (STI) | 0.19 | 0.21 | 0.37 | 0.25 | 0.35 | 0.23 | 0.21 | 0.22 | 0.19 |
| Yield index (YI) | 0.69 | 1.01 | 1.42 | 0.87 | 1.20 | 1.16 | 0.92 | 0.97 | 0.75 |
| Modified stress tolerance index-I (MSTI1) | 0.23 | 0.14 | 0.41 | 0.34 | 0.49 | 0.14 | 0.19 | 0.18 | 0.20 |
| Modified stress tolerance index- II (MSTI2) | 0.09 | 0.21 | 0.74 | 0.19 | 0.50 | 0.31 | 0.18 | 0.20 | 0.11 |
| Yield stability index (YSI) | 0.15 | 0.30 | 0.33 | 0.18 | 0.25 | 0.36 | 0.24 | 0.26 | 0.18 |
| Relative stress index (RSI) | 0.62 | 1.21 | 1.35 | 0.75 | 1.01 | 1.48 | 0.99 | 1.07 | 0.74 |
| Drought index (DI) | 0.11 | 0.30 | 0.47 | 0.16 | 0.30 | 0.42 | 0.22 | 0.25 | 0.14 |
| Stress/non-stress productivity index (SNPI) | 145.06 | 187.21 | 261.72 | 175.84 | 228.31 | 212.80 | 176.52 | 183.04 | 152.50 |
| Relative efficiency index (REI) | 0.77 | 0.84 | 1.50 | 1.02 | 1.42 | 0.92 | 0.86 | 0.88 | 0.77 |
| Mean relative performance (MRP) | 1.80 | 1.84 | 2.47 | 2.04 | 2.38 | 1.95 | 1.86 | 1.87 | 1.78 |
| Golden mean (GM) | 1.36 | 1.85 | 2.00 | 1.45 | 1.66 | 2.14 | 1.64 | 1.71 | 1.44 |
| Tolerance index (TOL) | 387.67 | 240.38 | 289.00 | 392.00 | 367.17 | 206.87 | 291.66 | 275.37 | 346.00 |
| Stress susceptibility Index (SSI) | 1.12 | 0.93 | 0.88 | 1.08 | 1.00 | 0.84 | 1.00 | 0.98 | 1.09 |
| Stress susceptibility percentage index (SSPI) | 0.47 | 0.29 | 0.35 | 0.48 | 0.45 | 0.25 | 0.35 | 0.33 | 0.42 |
| Yield reduction (YR) | 0.85 | 0.70 | 0.67 | 0.82 | 0.75 | 0.64 | 0.76 | 0.74 | 0.82 |
| Abiotic stress tolerance index (ATI) | 17048.80 | 11028.18 | 17731.07 | 19839.54 | 21944.46 | 9940.17 | 13584.70 | 12939.30 | 15264.15 |
| Mean productivity index (MPI) | 193.83 | 120.19 | 144.50 | 196.00 | 183.58 | 103.43 | 145.83 | 137.68 | 173.00 |
| Schnieder's stress susceptibility index (SSSI) | 0.09 | −0.05 | −0.09 | 0.06 | 0.00 | −0.12 | 0.00 | −0.02 | 0.06 |
| Sensitivity drought index (SDI) | 0.85 | 0.70 | 0.67 | 0.82 | 0.75 | 0.64 | 0.76 | 0.74 | 0.82 |

that low overlap existed between clusters, thus hard clustering methods were applied to construct the tanglegram (Fig. 5). Six hard clustering methods were compared using an agglomerative coefficient to choose the most accurate method for clustering the data, which were average, generalized average, single, weighted, complete, and Ward.

The valued of agglomerative coefficients were 0.76, 0.81, 0.53, 0.77, 0.85, and 0.88 respectively, under control treatment, whereas under 200 mM NaCl treatment, they

**Table 8** Rank of genotypes by 22 abiotic stress indices and shoot fresh weight under stress (Ys) and control (Yc) treatments as well as their average rank (AR).

| | Zhongsi 1084 | Gannong No. 2 | Gannong No. 4 | Shida No. 1 | C6 | C16 | C23 | C25 | C36 |
|---|---|---|---|---|---|---|---|---|---|
| Yield under normal condition (Yns) | 3 | 8 | 4 | 2 | 1 | 9 | 6 | 7 | 5 |
| Yield under stress condition (Ys) | 9 | 4 | 1 | 7 | 2 | 3 | 6 | 5 | 8 |
| Mean productivity (MP) | 4 | 8 | 2 | 3 | 1 | 9 | 6 | 7 | 5 |
| Geometric mean productivity (GMP) | 9 | 6 | 1 | 4 | 2 | 3 | 7 | 5 | 8 |
| Harmonic mean (HM) | 9 | 4 | 1 | 7 | 2 | 3 | 6 | 5 | 8 |
| Stress Tolerance Index (STI) | 9 | 7 | 1 | 3 | 2 | 4 | 6 | 5 | 8 |
| Yield index (YI) | 9 | 4 | 1 | 7 | 2 | 3 | 6 | 5 | 8 |
| Modified stress tolerance index-I (MSTI1) | 4 | 8 | 2 | 3 | 1 | 9 | 6 | 7 | 5 |
| Modified stress tolerance index- II (MSTI2) | 9 | 4 | 1 | 6 | 2 | 3 | 7 | 5 | 8 |
| Yield stability index (YSI) | 9 | 3 | 2 | 7 | 5 | 1 | 6 | 4 | 8 |
| Relative stress index (RSI) | 9 | 3 | 2 | 7 | 5 | 1 | 6 | 4 | 8 |
| Drought index (DI) | 9 | 3 | 1 | 7 | 4 | 2 | 6 | 5 | 8 |
| Stress/non-stress productivity index (SNPI) | 9 | 4 | 1 | 7 | 2 | 3 | 6 | 5 | 8 |
| Relative efficiency index (REI) | 9 | 7 | 1 | 3 | 2 | 4 | 6 | 5 | 8 |
| Mean relative performance (MRP) | 8 | 7 | 1 | 3 | 2 | 4 | 6 | 5 | 9 |
| Golden mean (GM) | 9 | 3 | 2 | 7 | 5 | 1 | 6 | 4 | 8 |
| Tolerance index (TOL) | 8 | 2 | 4 | 9 | 7 | 1 | 5 | 3 | 6 |
| Stress susceptibility Index (SSI) | 9 | 3 | 2 | 7 | 5 | 1 | 6 | 4 | 8 |
| Stress susceptibility percentage index (SSPI) | 8 | 2 | 4 | 9 | 7 | 1 | 5 | 3 | 6 |
| Yield reduction (YR) | 9 | 3 | 2 | 7 | 5 | 1 | 6 | 4 | 8 |
| Abiotic stress tolerance index (ATI) | 6 | 2 | 7 | 8 | 9 | 1 | 4 | 3 | 5 |
| Mean productivity index (MPI) | 8 | 2 | 4 | 9 | 7 | 1 | 5 | 3 | 6 |
| Schnieder's stress susceptibility index (SSSI) | 9 | 3 | 2 | 7 | 5 | 1 | 6 | 4 | 8 |
| Sensitivity drought index (SDI) | 9 | 3 | 2 | 7 | 5 | 1 | 6 | 4 | 8 |
| AR | 8.04 | 4.29 | 2.12 | 6.08 | 3.75 | 2.92 | 5.88 | 4.63 | 7.25 |

were 0.68, 0.72, 0.55, 0.73, 0.77, and 0.81 respectively. These results reveal that Ward's method had the highest coefficient compared to those of the other five methods under control and 200 mM NaCl treatments. Therefore, Ward's method was chosen to conduct cluster analysis. To identify the optimum number of clusters in the data, 30 internal validation indices were selected and screened (*Charrad et al., 2014*). As shown in Fig. 5, all genotypes were separated into two clusters under control and 200 mM NaCl treatment groups (Table 9). The structure of the clusters changed markedly when the genotypes were subjected to 200 mM NaCl treatment except for genotypes Gannong No. 4 and C6, which migrated from cluster 1 under control to cluster 2 under the saline treatment because they were more tolerant than the other members of their cluster.

Heatmaps elucidate the relationship between the genotypes and the studied traits based on standardized (scaled) data using a color scale under control and 200 mM NaCl treatments (Figs. 6 and 7). Before drawing the heatmap, the data were standardized by subtracting the mean from each value and dividing the obtained value by the standard deviation. Genotype C6 had the highest mean SFW and SDW in the control group, whereas

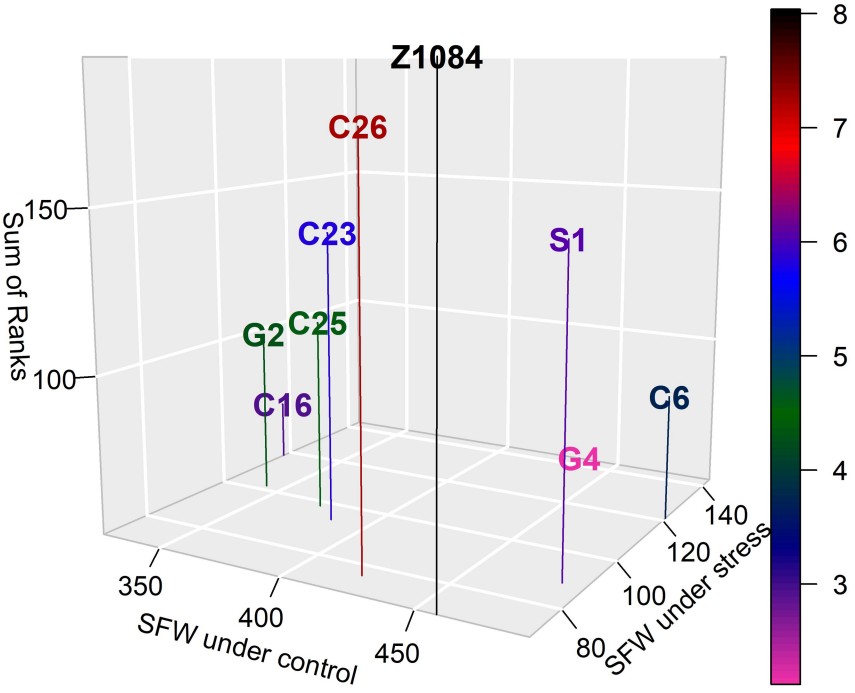

**Figure 4** Tolerance of genotypes according to the average rank of 22 abiotic stress indices (lower average rank indicates higher tolerance).

**Table 9** Average of the studied traits for the two clusters under normal and water stress conditions.

| Treatment | Control | | 200 mM | |
|---|---|---|---|---|
| Group | 1 | 2 | 1 | 2 |
| Germination rate (GR) | 4.28 | 4.65 | 1.35 | 1.54 |
| Germination vigor index (GVI) | 32.15 | 35.86 | 9.79 | 11.26 |
| Germination percentage (GP) | 86.89 | 88.71 | 27.17 | 31.13 |
| Mean daily germination (MDG) | 12.41 | 12.67 | 3.88 | 4.45 |
| Mean germination time (MGT) | 2.62 | 2.19 | 2.96 | 2.95 |
| Germination energy (GE) | 47.42 | 51.46 | 52.17 | 49.70 |
| Relative salt injury (RSI) | 0.00 | 0.00 | 69.46 | 65.02 |
| Seedling vigor index (SVI) | 14.32 | 14.73 | 0.55 | 0.81 |
| Shoot length (SL) | 9.84 | 9.80 | 1.56 | 2.19 |
| Root length (RL) | 6.49 | 6.73 | 0.48 | 0.48 |
| Root/shoot ratio (RSR) | 0.66 | 0.69 | 0.33 | 0.22 |
| Shoot fresh weight (SFW) | 410.15 | 415.33 | 87.97 | 127.66 |
| Shoot fresh weight (RFW) | 188.29 | 248.11 | 63.06 | 67.56 |
| Shoot dry weight (SDW) | 51.37 | 50.53 | 14.31 | 20.34 |
| Root dry weight (RDW) | 27.59 | 32.63 | 12.24 | 14.54 |

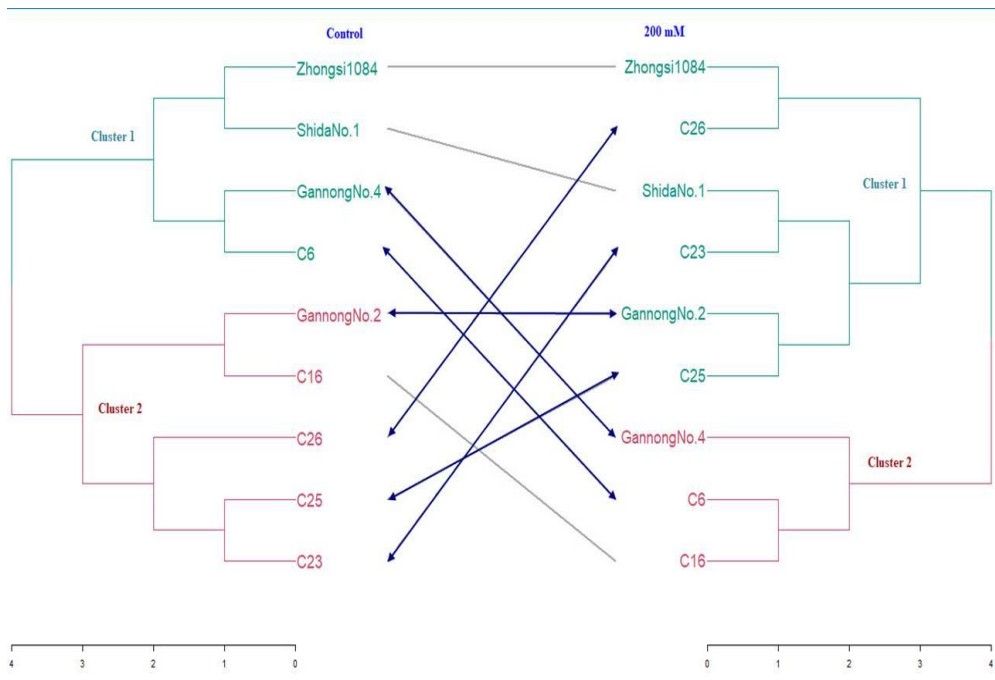

**Figure 5** Tanglegram showing results of cluster analysis based on Euclidian coefficient and Ward method under normal and water stress conditions.

genotype Gannong No. 4 had the highest mean SFW and SDW under the highest salinity treatment (200 mM). These results demonstrated that Gannong No. 4 was the most tolerant genotype. The lowest mean SFW and SDW under control treatment were observed in C16, whereas Zhongsi 1084 exhibited the lowest mean SFW and C26 had the lowest mean SDW under 200 mM NaCl treatment. Moreover, GP% of genotypes Gannong No. 4 and Gannong No. 2 was the highest and the lowest, respectively, under control treatment.

In contrast, the genotypes Zhongsi 1084 and Shida No. 1 were the highest and the lowest, respectively, under 200 mM. The genotype Zhongsi 1084 had higher values of germination traits under the highest salinity treatment. However, it had the lowest mean SFW, RFW, SL, and RSI. Gannong No. 4 had higher values of germination traits under control treatment. The heatmap does not reveal any association between germination traits and the tolerance indices of the genotypes, except for MGT, which was negatively associated with the tolerance of the genotypes.

## DISCUSSION

Soil salinity is a major environmental factor limiting crop growth and yield performance. Therefore, understanding these intricate responses is crucial for identifying specific salt-tolerant genotypes and optimizing agricultural practices in saline-prone regions. Therefore, the primary objective of this study was to identify salt-tolerant and sensitive triticale genotypes during the early seedling stage to assess their potential for salt tolerance. As
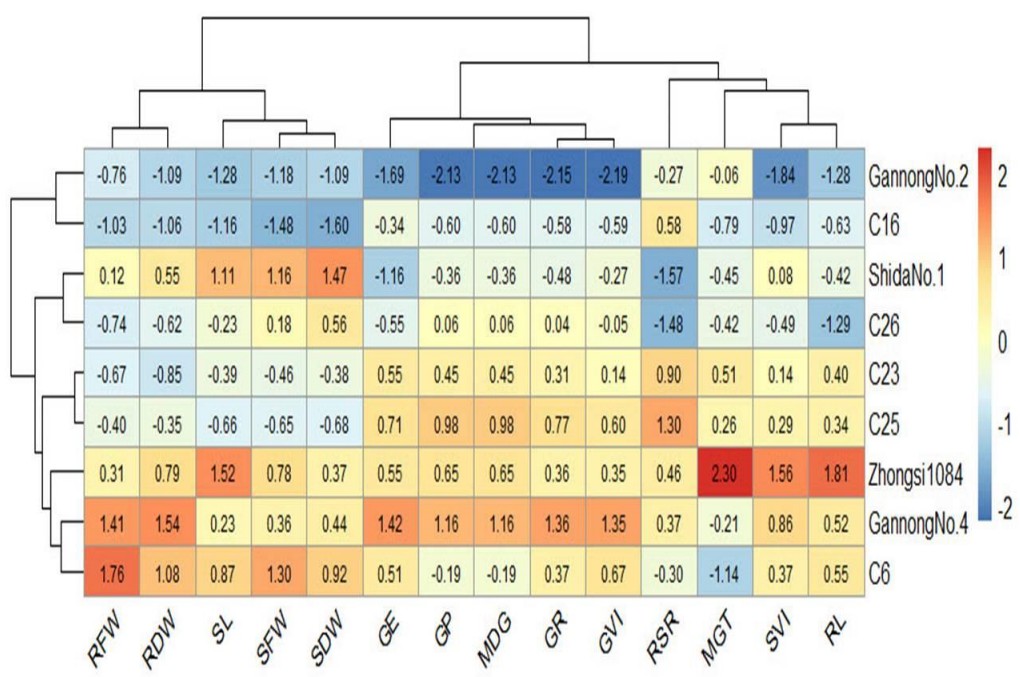

**Figure 6** Heatmap of the relationship between genotypes and the studied traits under control treatment.

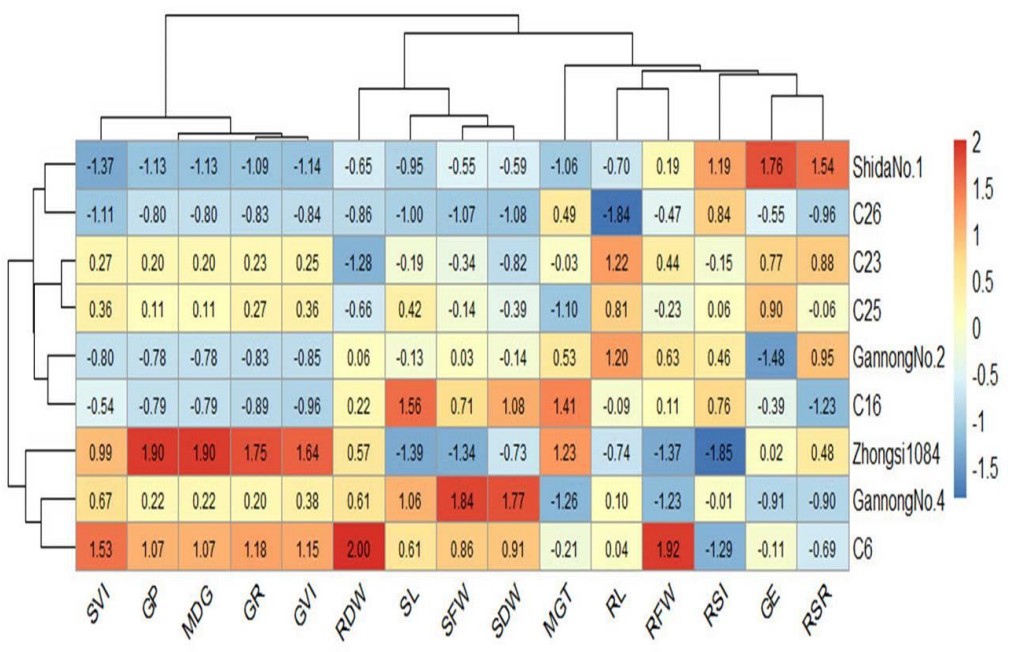

**Figure 7** Heatmap of the relationship between genotypes and the studied traits under 200 mM NaCl treatment.

previously noted, the impact of salinity on plant growth can differ significantly between plant species and even among different genotypes of the same species. Therefore, it is essential to monitor the genetic variability among genotypes (*Van Dijk et al., 2021*). This knowledge is crucial for enhancing salt tolerance in crops and improving their resilience to saline conditions. In consistent, our results indicated the differential responses of targeted genotypes to salinity stress. Based on germination traits, genotypes Zhongsi 1084, C6, C23, and C25 showed the highest salinity stress tolerance. Meanwhile, C6 and Gannong No. 4 were the most tolerant genotypes based on their seedling traits. In contrast, the germination traits of Gannong No. 2 and Shida No. 1 genotypes were the most sensitive genotypes. Genotype specific responses were also reported in the study of *Kandil et al. (2012)* who found that different wheat genotypes significantly varied in their response to salinity stress at GP%, GR, SVI, SL, RL, SFW, RFW, SDW, and RDW levels.

Different genotypes may demonstrate varying degrees of salinity stress tolerance at specific growth stages, necessitating careful selection and monitoring to ensure optimal performance under saline conditions. In this context, the effect of salinity stress is also associated with their growth stage (*Shannon, 1997*). Seed germination and seedling establishment are the most salt-sensitive stages of plants (*Ashraf & Foolad, 2005*). Salinity-induced ion toxicity, particularly elevated levels of $Na^+$ and $Cl^-$ ions, disrupts cellular processes, causing damage to plant tissues and hindering the normal growth and development of seedlings. *Atak et al. (2006)* found that high $Na^+$ accumulation induced germination inhibition. High salt levels in the soil can disrupt nutrient and water uptake, leading to stunted growth, leaf wilting, and other physiological stress symptoms (*Zhao et al., 2020*). In this regrad, the study of *Akgun, Kara & Altinda (2011)* found that GR, SL, RL and dry weights of the green parts and roots considerably decreased with increasing salt concentrations. *Kandil et al. (2012)* and *El kramany et al. (2009)* reported that with increasing salt concentrations, the average values of germination and seedling growth traits gradually reduced. *Francois et al. (1988)* also reported that at soil salinity of up 6.0 $dSm^{-1}$ and 20.5 $dSm^{-1}$ delayed the seed germination and reduced the final germination rate by 17%, respectively.

The salt tolerance index is commonly used to evaluate and rank the relative salt tolerance of different plant genotypes or varieties. Thus, correlating the salt tolerance index with these specific indices of germination and seedling growth shed the light on which traits contribute most to overall salt tolerance. This information can guide the selection and breeding of salt-tolerant plant varieties, ultimately leading to improved crop performance in saline-affected environments. In accordance with these results, *Alom et al. (2016)* reported that the salt tolerance index for seedling dry weight of wheat genotypes irrigated with saline water (15 $dSm^{-1}$) was positively correlated with salt tolerance indices GR, GVI, SL, and RL suggesting their role as selection criteria. *Aflaki et al. (2017)* investigated the effect of salinity on germination of different genotypes of wheat and found that MDG exhibited the highest correlation with GP%. In a previous study, PCA classified different genotypes of wheat and soybeans into three groups, *i.e.,* salt tolerant, moderately salt tolerant, and salt susceptible, based on the performance of these genotypes under different salt concentrations at the early seedling stage (*Saboora et al., 2006*; *Shelke et al., 2017*).

Overall, identifying which traits contribute most to overall salt tolerance, can guide the selection and breeding of salt-tolerant plant varieties, ultimately leading to improved crop performance in saline-affected environments.

## CONCLUSIONS

In the current study, the researchers observed that as the salt concentration increased, the average performance of most traits showed a gradual decrease. This indicates that higher salt levels negatively affected the performance of the plant traits under investigation. Correlating the salt tolerance index with these specific indices of germination and seedling growth and plant breeders can gain valuable insights into which traits contribute most to overall salt tolerance. By correlating the salt tolerance index with these specific indices such as GR, GVI, SL, and RL, we gain valuable insights into which traits contribute most to overall salt tolerance. Genotype Zhongsi 1084 exhibited the best germination performance. Line C6 and genotype Gannong No. 4 resulted in best performance for shoot and root length and fresh and root dry weight. PCA analysis grouped the most desirable genotypes (Gannong No.4 and C6) in clusters 1 and 2, whereas other genotypes were grouped into clusters 3 and 4. Overall, the identification of salt-tolerant traits in genotypes is crucial for addressing the challenges of salinity stress in agriculture and ensuring food security in the face of changing environmental conditions. The findings of our study will establish a basis for future research and offer valuable insights into the selection and development of salt-tolerant genotypes at early seedling stage.

### Funding
This research was funded by Princess Nourah bint Abdulrahman University Researchers Supporting Project number (PNURSP2024R402), Princess Nourah bint Abdulrahman University, Riyadh, Saudi Arabia. The funders had no role in study design, data collection and analysis, decision to publish, or preparation of the manuscript.

### Grant Disclosures
The following grant information was disclosed by the authors:
Princess Nourah bint Abdulrahman University Researchers: PNURSP2024R402.
Princess Nourah bint Abdulrahman University, Riyadh, Saudi Arabia.

### Competing Interests
The authors declare there are no competing interests.

### Author Contributions
- Ebrahim Ramadan conceived and designed the experiments, performed the experiments, authored or reviewed drafts of the article, and approved the final draft.
- Haytham A. Freeg conceived and designed the experiments, performed the experiments, authored or reviewed drafts of the article, and approved the final draft.

- Nagwa Shalaby conceived and designed the experiments, performed the experiments, authored or reviewed drafts of the article, and approved the final draft.
- Mosa S. Rizk conceived and designed the experiments, performed the experiments, authored or reviewed drafts of the article, and approved the final draft.
- Jun Ma conceived and designed the experiments, performed the experiments, authored or reviewed drafts of the article, and approved the final draft.
- Wenhua Du conceived and designed the experiments, performed the experiments, authored or reviewed drafts of the article, and approved the final draft.
- Omar M. Ibrahim conceived and designed the experiments, performed the experiments, analyzed the data, prepared figures and/or tables, authored or reviewed drafts of the article, and approved the final draft.
- Khairiah M. Alwutayd conceived and designed the experiments, authored or reviewed drafts of the article, and approved the final draft.
- Hamada AbdElgawad conceived and designed the experiments, authored or reviewed drafts of the article, and approved the final draft.
- Ick-Hyun Jo conceived and designed the experiments, authored or reviewed drafts of the article, and approved the final draft.
- Amira M. El-Tahan conceived and designed the experiments, performed the experiments, analyzed the data, prepared figures and/or tables, authored or reviewed drafts of the article, and approved the final draft.

## Data Availability

The raw data are available in the Supplemental Files.

## Supplemental Information

Supplemental information for this article can be found online at http://dx.doi.org/10.7717/peerj.16256#supplemental-information.

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
