# Peer review of "Response of nine triticale genotypes to different salt concentrations at the germination and early seedling stages"

_PeerJ, doi:10.7717/peerj.16256_

## Round 0.1 · original submission · Major Revisions

Dear Authors

Revise the manuscript as per the comments of the reviewers and resubmit for consideration.

Reviewer 1 ·

Basic reporting

Dear Editor
Thank you so much for giving me this opportunity to review the manuscript “#84031” entitled “Evaluation of some triticale genotypes under different salt concentrations at germination and early seedling stage”. It is a very well designed and well-constructed study. But the manuscript is not acceptable in its present format. The decision over the manuscript is “Major Revision”. All the required corrections are highlighted inside the manuscript with attached comment boxes. Authors are asked to go through all of them and correct them.
Comments:
1. Abstract: "had better performance for germination rate, germination vigor index,......" include some numerical data of results in this section as well.
2. Abstract: So, what is the concluding remark from this research work?? Need to mention it.
3. Keywords: Arrange all the keywords in alphabetical order. Try not to use abbreviated terms in the keyword section.
4. Introduction: "It is derived from the hybridization between .........." This particular statement needs a specific reference (updated-recent).
5. Introduction: Line number 33: "poorer soils"- means??
6. Introduction: Line number 38-39: Will the global population be stabilized in near future?? If yes than what is the assumed stabilized population number??
7. Introduction: Line number 38-40: Godfray et al., 2010; ray et al., 2013-these are very old references. Add some new updated references as the global population as well as statistics are also been changing.
8. Introduction: line number 73-77: After completion of a sentence, I think one should must add the "." wright?? And the novelty and the need of the research in the current day scenario needs to be highlighted more in the introduction section of the MS.
9. Materials and methods: Line number 80: Author need to add the altitude, latitude and longitude of the cultivation site.
10. Results and discussion: Results were included in an organized way, which is good for the manuscript but the discussion could have been better.
11. References: All the references must need to be in exact format that is required by the Journal.
12. Figure 1: Caption of Figure 1. should be in editable format. Resolution of figure must need to be increased.
13. Figure 2: Caption of Figure 1. should be in editable format. Resolution of figure must need to be increased.
14. Additionally, the revised version of the manuscript must be devoid of any grammatical or typical errors.

Experimental design

No comment

Validity of the findings

No comment

Additional comments

No comment

Annotated reviews are not available for download in order to protect the identity of reviewers who chose to remain anonymous.

·

Basic reporting

Comments to authors:
I have carefully reviewed the MS entitled “Evaluation of some triticale genotypes under different salt concentrations at germination and early seedling stage”. I have some major queries which should be addressed properly.
1. Were the environmental factors kept in mind or maintained to obtain the uniform results since environmental factors sometimes affect the growth and tolerance of the plant.
2. The results and section part is merely the representation of results with a minimum or negligible interpretation or discussion of the results. Incorporate more interpretation.
3. The MS as well as references are not formatted uniformly according to the Journal’s guidelines.
4. What is the novelty of the study? Mention in the text.

Experimental design

1. Why the cluster analysis was performed only on Shoot fresh weight (SFW) and Root fresh weight (RFW). Why the other traits were not included. Inclusion of more traits may create different clustering.

2. The experimental design for plantation is not clear and mentioned in material and methods section.

Validity of the findings

1. Tanglegram showed that the genotypes formed different clustering under normal and highest saline treatment. Write the rationale or hypothesis behind this change. The legend of figure 5 should be checked and corrected as it mentioned about water stress conditions.
2. The characteristics of the genotypes are missing in table 1.
3. In table 3 and 4, why the germination traits were separated by a line. Are both the germination traits, check and modify it accordingly.
4. In line 84, replace 0.0 mM with control.
5. In line 118, repetition of “using” should be removed.
6. In line 132, correct the genotype name
7. In line 139, meant daily germination should be corrected. It should be mean daily germination.
8. In line 364, complete the sentence “Complete, and ward.”

Additional comments

NA

·

Basic reporting

The manuscript has been well presented in context to the current scenario. Significant efforts were made while writing the MS. However, there are some major changes that need to be accomplished before it can be accepted for publication.
Line 28: The keyword doesn't sound appropriate in relation to the title and abstract. Try to add other convincing terms like- triticale, salinity, germination, seedling stage, PCA
Line 51-52: Line seems incomplete. reframe to make it meaningful
Line 68-71: line too lengthy and complex. Make it short and meaningful.
Line 74-75: When there is already a similar piece of work reported in this crop what is the significance and validity of the present work. Justify your MS with detailed explanation
Line 175-182: Do not just add the findings of previous authors. Make critical comparison of your data with those of the former and describe how different and significant is your current findings. The same goes for every part in the discussion section.
Line 410-411: The conclusion seems incomplete. Elaborate the significance of the present work.
Try to add more recent references and also the possible DOI numbers in the reference section.
Add some photographs, may be in supplementary file for phenotypic comparison of seed germination, especially the genotypes Zhongsi 10841048, C6, C23, and C25
The MS should be thoroughly revised with the help of fluent English speaker for proper framing of sentences.
Reference should be according to the journal's reference format throughout the MS. Some reference in reference section were missing the text. Add wherever applicable

Experimental design

Materials and methods well presented with sufficient information and can be replicated. All the required technical and ethical standards are followed in this section.

Validity of the findings

Data good enough to justify the findings and elaborated accordingly. Proper statistical tools were used for analysis. However, the discussion section is too short. Modified it to include more significant findings

Additional comments

Already included in basic reporting.

---

## Round 0.2 · Major Revisions

Still lot of corrections are required in the manuscript see the comments of reviewers and resubmit for consideration.

Reviewer 1 ·

Basic reporting

Dear Editor,
Thank you so much for providing me with the opportunity to re-review the manuscript “peerj-reviewing-84031-v1” entitled “Response of nine triticale genotypes to different salt concentrations at the germination and early seedling stages.” Authors have revised the manuscript very well and addressed almost all the previous comments in a good way. But, still for further proceeding of the MS author needs to make some minor correction as well. All the required corrections are highlighted inside the manuscript with attached comment boxes. Authors are asked to go through all of the and correct them. The decision over the manuscript is “Minor revisions”.
Comments:
1. Introduction: Page 9, Line number 81: This nay??.... What’s the meaning of nay?? Was it a typographical error? If yes resolve it.
2. Results: Page number 14, Line number 201: Avoid using words such as you, I, we, they, etc.
3. Page 16, Line number 247: GP %: Write and present all the GP% in similar fashion throughout the whole manuscript, including the difference in spaces also.
4. Page number 18, line number 289, 290: 200 Mm, 120 Mm etc., need to be written in its appropriate format, such as 200 mM, 120 mM etc. Correct these types of mistakes throughout the whole manuscript.
5. Page 19, Line number 318: Do not write NaCl as NaCL.. Correct it throughout the manuscript.
6. Discussion: Page 23, Line number 394: Headings must need to be in accordance to the Journal format only.
7. Page 24, Line number 420, 421: From this paragraph onwards, there is a problem in MS scale. Resolve it.
8. Page 25, Line number 439: (cm)shoot: spacing issue is there. Resolve it.
9. Page 25, Line number 445: "germination." : spacing issue is there. Resolve it.
10. References: Double check the references for any grammatical, typical errors. And volume, issue, page numbers and publication years must need to be presented appropriately, as the Journal demands.

Experimental design

NA

Validity of the findings

NA

Additional comments

NA

Annotated reviews are not available for download in order to protect the identity of reviewers who chose to remain anonymous.

·

Basic reporting

1. As asked to write about the significance of the present work in conclusion section was not incorporated. Only the results were observed in conclusion which needs modification.
2. The discussion was again not up to the expectation, only findings of the earlier studies were presented.
3. References were not uniformly formatted according to journal's guidelines.
4. The other queries have been addressed accordingly.

Experimental design

NA

Validity of the findings

NA

Additional comments

NA

---

## Round 0.3 · Minor Revisions

Please see the comments and resubmit the manuscript.

Reviewer 1 ·

Basic reporting

Dear Editor,
Thank you so much for providing me with the opportunity to re-review the manuscript “peerj-reviewing-84031-v2” entitled “Response of nine triticale genotypes to different salt concentrations at the germination and early seedling stages.” Authors have revised the manuscript very well and addressed all the previous comments in a good way. The manuscript now can be processed further for publication. The decision over the manuscript is “Accept”.

Experimental design

NA

Validity of the findings

NA

Additional comments

NA

·

Basic reporting

The authors have addressed all the corrections provided.

Experimental design

NA

Validity of the findings

NA

Additional comments

NA

·

Basic reporting

I can see the authors have made sufficient changes in the MS. However, all the previous comments were not properly addressed in the rebuttal letter, some of which I have mentioned below-
line 74-75: When there is already a similar piece of work reported in this crop what is the significance and validity of the present work. Justify your MS with detailed explanation
Line 175-182: Do not just add the findings of previous authors. Make critical comparison of your data with those of the former and describe how different and significant is your current findings.
Try to add more recent references and also the possible DOI numbers in the reference section.
Add some photographs, may be in supplementary file for phenotypic comparison of seed germination, especially the genotypes Zhongsi 10841048, C6, C23, and C25

Also some lines are incomplete in the present MS which I have marked in the pdf file. The comments addressed in the new PDF file also requires changes.

Experimental design

N/A

Validity of the findings

N/A

Additional comments

N/A

---

## Round 0.4 · accepted · Accept

All comments has been resolved properly.